# Mediating Effects of Specific Types of Coping Styles on the Relationship between Childhood Maltreatment and Depressive Symptoms among Chinese Undergraduates: The Role of Sex

**DOI:** 10.3390/ijerph17093120

**Published:** 2020-04-30

**Authors:** Xianbing Song, Shanshan Wang, Rui Wang, Huiqiong Xu, Zhicheng Jiang, Shuqin Li, Shichen Zhang, Yuhui Wan

**Affiliations:** 1Department of Human Anatomy, Histology & Embryology, Anhui Medical College, Hefei 230032, China; ayzjc2008@163.com; 2Department of Maternal, Child & Adolescent Health, School of Public Health, Anhui Medical University, Hefei 230032, China; 13625510589@163.com (S.W.); xhqxuhuiqiong@163.com (H.X.); 18895672886@163.com (Z.J.); LiShuqin1018@163.com (S.L.); zhangshichen@ahmu.edu.cn (S.Z.); 3Anhui Provincial Key Laboratory of Population Health & Aristogenics, Anhui Medical University, Hefei 230032, China; 4Information Technology Office, Anqing Medical College, Anqing 246052, China; aqyzwr@126.com

**Keywords:** childhood maltreatment, coping style, depressive symptoms, undergraduates, sex

## Abstract

Although childhood maltreatment is known to be associated with depressive symptoms, few studies have investigated the mediating effect of different types of coping styles on this association. It is unknown whether the impacts vary by sex. We investigated the mediating effects of different coping styles on the relationship between childhood maltreatment and depressive symptoms among Chinese undergraduates, as well as the role of sex in the mediated pathways. A total of 7643 college freshmen and sophomores (5665 females, 1978 males; 4215 freshmen, 3428 sophomores; mean age, 19.67 years) from two colleges in China completed a standard questionnaire on the details of childhood maltreatment, depressive symptoms, and coping styles. Childhood maltreatment was significantly correlated with all coping styles and depressive symptoms studied (*p* < 0.001). Problem solving, self-blame, help seeking, problem avoidance, and rationalization mediated the relationship between childhood maltreatment and depressive symptoms. The estimated ratio of the effect of childhood maltreatment on the occurrence of depressive symptoms can be explained by the mediation of problem solving, self-blame, help seeking, problem avoidance, and rationalization, which accounted for 15.1%, 25.6%, 7.4%, 1.6% and 1.6% of the total effect, respectively. Sex differences were found to have mediating effects on coping styles in terms of the relationship between childhood maltreatment and depressive symptoms. The findings illustrate the need to focus on coping styles and to employ sex-specific methods to effectively help college students reduce depressive symptoms associated with childhood maltreatment.

## 1. Introduction

Childhood maltreatment is behavior toward a person under 18 years of age, including abuse or neglect, that results in actual or potential harm to the child’s health, survival, development or dignity and is perpetrated by a person of responsibility, trust, or power in that child’s life [1,2]. Exposure to childhood maltreatment is associated with persistent symptoms of depression throughout a person’s lifespan [3,4]. For example, a meta-analysis examining the relationship between childhood maltreatment and adult depression demonstrated that nearly half of patients with depression reported a history of childhood maltreatment, and maltreatment exposure contributes to an increased risk of developing depression in adulthood [5]. A survey of 51,945 adults suggested that childhood adversity (child abuse and neglect) is an important influencing factor in the incidence of mental illness [6]. Research by Vallati et al. also showed that high levels of emotional maltreatment and/or sexual maltreatment were significantly associated with severe depressive symptoms [3]. Although evidence has shown a relationship between childhood maltreatment and depressive symptoms, the mediated psychological mechanism by which these factors are linked has not been established.

It has been suggested that coping styles are involved in this pathway. Coping, as defined by Lazarus and Folkman [7], is “constantly changing cognitive and behavioral efforts to manage specific external and/or internal requirements that are identified as taxing or exceeding the resources of the person”. A healthy environment is a safe environment that enables children to develop emotionally and socially, thereby helping them to pay attention to the environment, and learn what they can expect from it and respond to it [8]. Research has indicated that children who live in stressful family environments (children who are raised in poverty or maltreated by their parents) are more likely to adopt avoidant emotion-focused strategies (i.e., reducing negative emotional responses due to stressors, but the role of solving the actual pressure source is small) than those who do not [9]. In particular, abused children often consider their environment to be threatening and unpredictable, and they have no ability to change it [10]. For example, Gipple et al. revealed that childhood abuse experiences (sexual abuse, physical abuse, and negative home environment) are associated with a decrease in the frequency of problem-focused coping (including planning and active coping), which has been defined as behavioral and cognitive efforts to change or remove a stressor [11]. However, studies about the link between childhood maltreatment and specific coping styles remain rare. Coping styles may have an impact on the presence of depressive symptoms, and some studies have shown a link between coping and psychological adjustment [12,13]. Despite the fact that emotion-focused coping strategies (including venting, positive reappraisal, rumination, and self-blame) may have advantages in coping with certain stressors, findings in the literature often suggest that emotion-focused coping can predict higher levels of psychopathology and functional impairment [14,15]. In addition, emotion-focused coping strategies are associated with higher levels of anxiety, depression, and distress in both nonclinical [16] and clinical samples [15,17]. Therefore, the evidence indicates that the tendency to use specific types of coping styles is likely to arise from childhood maltreatment and that these styles are associated with depressive symptoms, which highlights that specific types of coping styles may be mediators.

Despite some research on sex differences in coping styles, females are more likely to use emotion-focused coping strategies, and males are more likely to be involved in problem-focused coping strategies [18]. Sex differences exist in mental health, where women are more susceptible than men to internalizing disorders such as depression [19]. The National Comorbidity Survey Replication describes the prevalence of mental illness in the general population of the U.S.; 46 million women (29%) suffer from depression throughout their lives, compared to 28 million men (18%) [20]. Previous studies have also reported sex differences in the prevalence of child maltreatment, with girls (13.0% (95% CI, 12.9–13.0%)) reporting higher rates of maltreatment than boys (12.0% (95% CI, 12.0–12.1%)) [21]. Furthermore, Matud et al. have speculated that differences in the way women cope with stress could be related to their higher levels of psychological distress, symptoms of depression and anxiety compared with men [18]. However, no studies have examined the impact of sex on the mediating effect of specific types of coping styles on the relationship between childhood maltreatment and depressive symptoms among college students.

Hence, the current study aims to identify the relationships between childhood maltreatment, coping styles, and depressive symptoms and investigate the possible mediating roles of different types of coping styles in the relationship between childhood maltreatment and depressive symptoms among Chinese undergraduates and the sex differences in the mediated pathways. Based on previous research, the following hypotheses were proposed: (1) there are significant correlations between childhood maltreatment, coping styles and depressive symptoms; (2) specific types of coping styles mediate the relationship between childhood maltreatment and depressive symptoms; and (3) coping styles, as mediating roles, vary by sex in the association between childhood maltreatment and depressive symptoms.

## 2. Methods

### 2.1. Sample and Procedures

From October to December 2019, we carried out a cross-sectional survey with a sample of 8198 students with a mean age of 19.67 years (SD 1.11) from two colleges in Hefei and Anqing City in Anhui Province, China. In each school, all freshmen and sophomores were selected for inclusion within the survey. The students were asked to complete an anonymous questionnaire. In the present study, investigators entered schools and classrooms to conduct surveys and did not occupy students’ class hours. The investigation was located in the classroom. Teachers were invited to take charged for maintaining discipline, but not to participate in the survey. The teachers sent the link of the electronic questionnaire to each student’s mobile phone, and then the students began to complete the questionnaire. The investigators introduced the objective of the survey and the questions to be aware of when filling out questionnaires, and emphasized the principles of anonymity and confidentiality participation. Investigators answered questions from students on the spot. Due to an unwillingness to respond to the questionnaire, absence from school, high levels of missing data (questionnaires with missing values >5% were eliminated), or obviously fictitious responses, 555 (6.8%) participants were excluded from the study. Thus, the data from 7643 (93.2%) participants were analyzed. More female students than male students participated in the study (65.5% vs. 34.5%), and the respondents were distributed in the Anhui Medical College (4206 students) and Anqing Medical College (3437 students). Among all participants, 63.2% came from rural areas and 36.8% from urban areas. Approximately three-quarters (78.5%) reported that they were the only child and 21.5% had at least one sibling.

The study design and data collection procedures were both approved by the Ethics Committee of Anhui Medical University (20170290). Informed consent was obtained from all of the students.

### 2.2. Measures

#### 2.2.1. Depressive Symptoms

Depressive symptoms were evaluated using the Center for Epidemiologic Studies Depression (CES-D) scale [22]; the Chinese version of the questionnaire was widely used in Chinese populations and showed good reliability and validity [23]. The CES-D is a 20-item self-administered questionnaire in which participants evaluate depressive symptoms experienced during the previous week on a one (“rarely or none of the time”) to four (“most or all of the time”) scale, resulting in a global score ranging from zero to 80. The internal consistency of this scale is generally high (Cronbach’s α = 0.822 in the present study). The higher the score on the questionnaire, the higher the severity of the symptoms of depression experienced.

#### 2.2.2. Coping Styles

Coping styles were assessed using the coping style questionnaire (CSQ), which was created based on the Chinese culture and language [24]. The CSQ consists of 62 questions in six dimensions, as follows: problem solving (e.g., “I can learn from myself or others to cope with difficulties”), self-blame (e.g., “I often blame myself”), help seeking (e.g., “I will ask for someone’s help to deal with difficulties”), fantasizing (e.g., “I often fantasize about unrealistic things to eliminate troubles”), problem avoidance (e.g., “I do not think much about a situation that makes me upset”), and rationalization (e.g., “conflict with another is due to his or her bad personality”). The participants were asked whether they preferred to use each of these coping strategies (“no” = 0, “yes” = 1). The total scores of each subscale were calculated. The CSQ has been widely used in study on coping styles among Chinese undergraduates [24,25]. In the current research, Cronbach’s α values for the six subscales were as follows: problem solving, 0.727; self-blame, 0.826; help seeking, 0.670; fantasizing, 0.717; problem avoidance, 0.732; and rationalizing, 0.637.

#### 2.2.3. Childhood Maltreatment

Childhood maltreatment was evaluated using the Child Trauma Questionnaire (CTQ) [26], a widely used 28-item measure that assesses five different forms of childhood trauma (including physical abuse, sexual abuse, emotional abuse, physical neglect, and emotional neglect). The CTQ has been translated and validated in Chinese [27]. The participants were asked if they had experienced childhood abuse and neglect before they had reached the age of 16 years. Response scores ranged from 1 = “never true,” to 5 = “very often true.” A higher score reflects more serious levels of childhood maltreatment. The Cronbach’s α coefficient for the childhood maltreatment scale was 0.745. The Cronbach’s α coefficients for the emotional abuse, physical abuse, sexual abuse, emotional neglect, and physical neglect scales in this study were 0.749, 0.774, 0.726, 0.788, and 0.680, respectively.

#### 2.2.4. Covariates

This study also controlled for participants’ age, urban/rural status, school, only child status (whether the participants had any siblings), parental educational level (less than junior middle school, junior middle school, senior middle school, college or higher), and the perceived economic status of the family (poor, moderate, or good), all of which may affect the experience of depressive symptoms [28].

### 2.3. Analyses

All data were analyzed using SPSS (version 23.0, SPSS Inc., Chicago, IL, USA). First, descriptive statistics were used to describe childhood maltreatment, coping styles and depressive symptoms. Next, we conducted Spearman correlations to test the associations among childhood maltreatment, coping styles and depressive symptoms. In addition, a multiple linear regression analysis was carried out among undergraduates to evaluate the relationships between the dependent variable of depressive symptoms and childhood maltreatment and coping styles as explanatory variables. Then, the PROCESS program of mediation (model 4) was used to perform a multiple mediation analysis [29,30]. To examine any possible mechanisms underlying the significant factors of depressive symptoms and childhood maltreatment as described above, we tested the role of problem solving, self-blame, help seeking, fantasizing, problem avoidance, and rationalization as mediators. A multiple mediation analysis was used, which examines multiple variables and their indirect effects simultaneously [31]. This approach uses bootstrapping to estimate all of the parameters. The mediating effect was tested using a bootstrap estimation approach with 5000 repetitions. When the 95% CI did not contain zero, the indirect effect was considered significant. Finally, we explored whether different types of coping styles mediated the relationship between childhood maltreatment and depressive symptoms among males and females.

## 3. Results

Descriptive statistics related to childhood maltreatment, coping styles and depressive symptom development are presented in Table 1. Bivariate correlations between the main variables are presented in Table 2. Spearman correlation analyses revealed significant associations between childhood maltreatment and depressive symptoms (*p* < 0.001), problem solving (*p* < 0.001), self-blame (*p* < 0.001), help seeking (*p* < 0.001), fantasizing (*p* < 0.001), problem avoidance (*p* < 0.001), and rationalization (*p* < 0.001). Compared with other types of coping styles, childhood maltreatment was most strongly associated with self-blame (*r* = 0.29, *p* < 0.001). In addition, the analyses revealed significant correlations between depressive symptoms and problem solving (*p* < 0.001), self-blame (*p* < 0.001), help seeking (*p* < 0.001), fantasizing (*p* < 0.001), problem avoidance (*p* < 0.001), and rationalization (*p* < 0.001). Compared with other types of coping styles, self-blame was most strongly associated with depressive symptoms (*r* = 0.59, *p* < 0.001).

The results of the regression analysis indicate that self-blame interpreted the foremost variance in depressive symptoms (*β* = 0.375, *p* < 0.001) (childhood maltreatment: *β* = 0.218, *p* < 0.001; problem solving: *β* = −0.232, *p* < 0.001; help seeking: *β* = −0.121, *p* < 0.001; fantasizing: *β* = 0.011, *p* = 0.364; problem avoidance: *β* = 0.032, *p* < 0.05; rationalization: *β* = 0.036, *p* < 0.05) (Table 3). We also performed a single factor regression analysis between childhood maltreatment and depressive symptoms, which showed that childhood maltreatment was significantly associated with depressive symptoms (*β* = 0.454, *p* < 0.001; *F* = 1983.075, *p <* 0.001; *R*^2^ = 0.206).

Subsequently, the standardized indirect effects of the mediating variable included in the model were calculated, and we tested whether the effect was statistically significant. Table 4 shows that after adjusting for all covariates (including age, gender, urban/rural, school, only child status, parents’ education level, and family economic status) in model 2, fantasizing did not mediate the relationship between childhood maltreatment and depressive symptoms. The indirect effects of problem solving, self-blame, help seeking, problem avoidance, and rationalization accounted for 15.1%, 25.6%, 7.4%, 1.6% and 1.6% of the total effect in the total sample, respectively. In model 3, after adjusting for participants’ age, urban/rural status, school, only child status, parents’ education level, and family economic status, the effects of problem solving, self-blame, help seeking, problem avoidance, and rationalization accounted for 13.9%, 24.8%, 7.9%, 2.0% and 1.8% of the total effect among females, respectively. However, in the males, only problem solving, self-blame, and help seeking mediated the relationship between childhood maltreatment and depressive symptoms, and the indirect effects of these variables accounted for 18.5%, 28.5%, and 6.0%, respectively, of the total effect. Self-blame had a stronger mediating effect than any other type of coping style. Problem solving, self-blame, help seeking, problem avoidance, and rationalization mediated the relationship between different types of childhood maltreatment and depressive symptoms. Sex differences were also found in the mediating effects of coping styles on the relationship between all types of childhood maltreatment and depressive symptoms. Within the relationship between different types of childhood maltreatment and depressive symptoms, the mediating effects of self-blame were always larger than those of any other types of coping style (refer to attached Appendix A).

## 4. Discussion

This study explored the relationship between childhood maltreatment and depressive symptoms, as well as the mediating role of coping styles, among Chinese undergraduates. As expected, childhood maltreatment was positively related to depressive symptoms and can also affect depressive symptoms indirectly via problem solving, self-blame, help seeking, problem avoidance, and rationalization. Moreover, the mediating effect of problem avoidance and rationalization on the relationship between childhood maltreatment and depressive symptoms was found among females but not among males.

### 4.1. Correlations between Childhood Maltreatment, Coping Styles, and Depressive Symptoms

The first objective of this study was to identify the relationships between childhood maltreatment, coping styles, and depressive symptoms. The significant correlations found between the different variables, usually in the direction we expected. Childhood maltreatment had a positive correlation with the development of depressive symptoms among undergraduates, i.e., the more childhood maltreatment there is, the more psychological symptoms there are. These data coincide with those of previous findings [3,4,5]. Likewise, the positive relationships between childhood maltreatment and self-blame, fantasizing, problem avoidance, and rationalization were confirmed. Research by Milojevich et al. [32] demonstrated that greater exposure to physical and/or sexual abuse predicted a greater use of avoidant strategies in adolescence. Childhood maltreatment was significantly and negatively associated with problem solving and help seeking. Min et al. [33] confirmed that adolescents who experienced child maltreatment reported less dominant use of coping styles such as problem solving, emotional regulation and cognitive restructuring, and more dominant use of denial, avoidance and inaction than adolescents who did not experience child maltreatment. Compared with the other types of coping styles, self-blame was most strongly associated with childhood maltreatment. This finding may be consistent with the research purporting that self-blame is an internal ascription and a cognitive process. Individuals ascribe the occurrence of negative events to themselves; in the case of abuse (such as childhood sexual abuse), some victims ascribe the abuse to an internal cause [34].

In addition to the link with childhood maltreatment, depressive symptoms were positively associated with self-blame, fantasizing, problem avoidance, and rationalization and negatively associated with problem solving and help seeking, corroborating the conclusions of the study [25], which showed that depression was negatively associated with problem solving and help seeking but positively associated with self-blame, fantasizing, avoidance, and rationalization. Compared with other types of coping styles, self-blame was most significantly associated with depressive symptoms. Studies have evaluated nonclinical and clinical samples, indicating that emotion-focused coping strategies (such as self-blame) are associated with high levels of anxiety, depression, and other mental illnesses [15,16].

### 4.2. The Mediating Role of Coping Style and Sex Differences

However, the analysis of depressive symptoms should be considered from a more sophisticated angle, rather than simply focusing on bivariate relationships. These relationships may be affected by another variable, such as some mediators [35], which may help us understand more about how or why two variables are related at all. In fact, another purpose of this research was to estimate the possible mediating role of various types of coping styles in the link between childhood maltreatment and depressive symptoms among Chinese undergraduates. The results derived from the multiple mediation model in the present study showed the apparent mediating roles of problem solving, self-blame, help seeking, problem avoidance, and rationalization in the relationship between childhood maltreatment and depressive symptoms. Littleton et al. confirmed that disengagement coping strategies related to traumatic events (including avoidance, denial and social withdrawal) may increase the incidence of psychological distress [36]. The review by Whiffen et al. pointed that shame, self-blame, and avoidant coping strategies mediated the relationship between childhood sexual abuse and psychological problems. However, few studies have considered incorporating these factors (self-blame, shame, and coping strategies) into the same model to analyze their impact on depressive symptoms [37]. The theoretical models proposed by Nusslock et al. [38,39] showed that the biological and psychosocial changes caused by childhood maltreatment are predictors of many negative outcomes in adulthood (e.g., health status, drug abuse, psychological status). These biopsychosocial models show that, in particular, avoidant emotion-focused coping strategies parallel and interact with other biological (e.g., impaired immune function) and psychosocial (e.g., problematic health behaviors) pathways that affect adult diseases. One explanation for this is that childhood maltreatment may increase the sensitivity of brain regions involved in stress responses, inhibitory control, and reward responses [38]. These neurobiological and psychological changes, in turn, affect the cognitive assessment of threats and the response to perceived threats. Therefore, people who are exposed to childhood maltreatment are more likely to experience stressful situations, but less likely to develop effective coping styles, which might result in more psychological problems.

Self-blame was found to exert a larger mediating effect than other types of coping styles for both females and males. Childhood maltreatment appears to have an indirect relationship with depressive symptoms, which are more likely to be expressed via a self-blame coping style. A study based on health survey in the United States revealed that individuals who experienced childhood maltreatment use more negative coping strategies (the emotion-focused coping style and avoidant coping style) than those who did not, leading to an elevated risk of mental health issues [40]. Mennin et al. also demonstrated that avoidant and emotion-focused coping strategies could promote anxiety [41,42]. In exploring the mediating role of coping styles in the association between childhood maltreatment and depressive symptoms, existing research has not distinguished coping styles in detail, but identifying the coping styles that are most relevant to the reduction in depressive symptoms among Chinese undergraduates in greater depth is imperative, in order that interventions can be specifically focused on the most relevant coping styles.

In the present study, problem solving, self-blame, help seeking, problem avoidance, and rationalization mediated the relationship between childhood maltreatment and depressive symptoms among females, while the mediating effects of problem avoidance and rationalization were not found for males. This finding may be interpreted within the cultural context of China. In China, different sexes play different roles in family and society. Compared to females, males are always expected to shoulder more responsibilities and pressure from family and society [43]; Chinese males are more self-sufficient and independent, and when confronted with childhood maltreatment experiences, females are more likely to develop problem avoidance and rationalization coping styles than males, who may be more inclined to engage in depressive symptoms. Additionally, research by Horwitz et al. [44] demonstrated that females are more likely to engage in emotion-focused coping strategies (i.e., changing how people feel about a situation) than males, and males are more likely to engage in problem-focused coping strategies (i.e., taking action and changing the situation) than females.

Childhood maltreatment may influence an individual’s view of himself/herself [45]; however, coping styles are prone to constant change, and their perspectives could be modulated in a positive way, given the right tools. Thus, developing problem solving and help seeking coping styles while minimizing self-blame, problem avoidance, and rationalization coping styles—and, particularly, limiting self-blame—would prevent the occurrence of more depressive symptoms resulting from the exposure to childhood maltreatment among the high-risk undergraduate population. This population is an important target for intervention in addition to trauma therapy, where appropriate. Moreover, the preliminary exploration of the factors affecting the association between childhood maltreatment and depressive symptoms in this study will hopefully provide a reference for future cohort studies.

### 4.3. Strengths and Limitations

The main contribution of this study, beyond the intrinsic importance of studying childhood maltreatment in relation to depressive symptoms among undergraduates, lies in its revelation of the roles of specific types of coping styles in the relationship between these two factors. In addition, the large sample coverage of both urban and rural areas in China indicates the better representation of the study. However, several limitations should also be noted when interpreting these results. First, the cross-sectional design of this study may induce vagueness in the temporal ordering of the variables. Second, the background of participants in this study were medical university environments; as such, the findings did not represent people absent from university, a point worth noting, since childhood maltreatment and depressive symptoms are more prevalent among individuals with lower educational accomplishments and a worse economic situation. Thus, the effects uncovered by the current study may be underestimated. Third, given that all participants were from one city in mainland China, the degree to which one can generalize these findings to young adults in other countries or cultures needs further elaboration. Fourth, childhood maltreatment was assessed by a retrospective questionnaire, and thus recall bias cannot be avoided in this study. Lastly, the fact that depressive symptoms were investigated using a screening questionnaire instead of a specific psychometric instrument should be considered.

## 5. Conclusions

The mediating roles of problem solving, self-blame, help seeking, problem avoidance, and rationalization were all observed in the relationship between childhood maltreatment and depressive symptoms. Sex differences were found in the mediating effects of coping styles on the relationship between childhood maltreatment and depressive symptoms. Hopefully, these findings will have implications for the ways in which childhood maltreatment and coping styles, especially regarding self-blame, are incorporated into prevention and intervention programs to address depressive symptoms among young adults.

## Figures and Tables

**Table 1 ijerph-17-03120-t001:** Descriptive statistics for variables.

Variables	Minimum	Maximum	Mean	SD
Emotional abuse	5	25	6.79	2.48
Physical abuse	5	25	5.58	1.47
Sexual abuse	5	25	5.24	0.92
Emotional neglect	5	25	10.72	5.00
Physical neglect	5	25	8.06	2.89
Childhood maltreatment	25	98	36.39	9.02
Problem solving	0	12	8.96	2.54
Self-blame	0	10	3.26	2.80
Help seeking	0	10	5.96	2.40
Fantasy	0	10	4.48	2.37
Problem avoidance	0	11	4.38	2.65
Rationalization	0	11	4.49	2.20
Depressive symptoms	20	80	32.30	9.87

**Table 2 ijerph-17-03120-t002:** Correlations among childhood maltreatment, coping style, and depressive symptoms in Chinese adolescents.

Variable	1	2	3	4	5	6	7	8	9	10	11	12	13
1. Emotional abuse	1	0.45 **	0.25 **	0.24 **	0.17 **	0.49 **	−0.22 **	0.34 **	−0.21 **	0.30 **	0.29 **	0.23 **	0.37 **
2. Physical abuse		1	0.24 **	0.14 **	0.11 **	0.35 **	−0.13 **	0.21 **	−0.14 **	0.18 **	0.18 **	0.15 **	0.21 **
3. Sexual abuse			1	0.11 **	0.10 **	0.24 **	−0.10 **	0.17 **	−0.11 **	0.14 **	0.15 **	0.13 **	0.19 **
4. Emotional neglect				1	0.56 **	0.88 **	−0.23 **	0.20 **	−0.23 **	0.11 **	0.17 **	0.12 **	0.35 **
5. Physical neglect					1	0.75 **	−0.17 **	0.17 **	−0.19 **	0.08 **	0.15 **	0.11 **	0.30 **
6. Childhood maltreatment						1	−0.27 **	0.29 **	−0.28 **	0.20 **	0.25 **	0.20 **	0.44 **
7. Problem solving							1	−0.27 **	0.46 **	−0.15 **	−0.20 **	0.01	−0.44 **
8. Self–blame								1	−0.31 **	0.67 **	0.67 **	0.56 **	0.59 **
9. Help seeking									1	−0.15 **	−0.22 **	−0.08 **	−0.41 **
10. Fantasy										1	0.69 **	0.58 **	0.42 **
11. Problem avoidance											1	0.64 **	0.46 **
12. Rationalization												1	0.35 **
13. Depressive symptoms													1

** *p* < 0.001.

**Table 3 ijerph-17-03120-t003:** Multiple regression analysis of childhood maltreatment, coping styles predicting depressive symptoms in adolescents.

Variable	*B*	*SE*	*β*	*p*
Depressive symptoms	*F* = 1150.453 **; *R*^2^ = 0.513			
Childhood maltreatment	0.239	0.010	0.218	0.000
Problem solving	−0.902	0.037	−0.232	0.000
Self-blame	1.324	0.045	0.375	0.000
Help seeking	−0.498	0.039	−0.121	0.000
Fantasy	0.047	0.052	0.011	0.364
Problem avoidance	0.111	0.050	0.030	0.026
Rationalization	0.162	0.054	0.036	0.003

** *p* < 0.001.

**Table 4 ijerph-17-03120-t004:** Mediating role of different types of coping styles in the association between childhood maltreatment and depressive symptoms.

Mediator	Model	a	b	Direct Effect	Boot CI		Indirect Effect	Boot CI		Mediation Ratio, %
				c′	LLCI	ULCI	(a × b)	LLCI	ULCI	a × b/(a × b + c′)
**Total**										
Problem solving	1	−0.080 **	−0.902 **	0.239 **	0.221	0.258	0.072	0.063	0.081	14.5
	2	−0.081 **	−0.913 **	0.233 **	0.215	0.252	0.074	0.065	0.083	15.1
Self-blame	1	0.098 **	1.324 **	0.239 **	0.221	0.258	0.130	0.117	0.143	26.1
	2	0.096 **	1.307 **	0.233 **	0.215	0.252	0.125	0.113	0.138	25.6
Help seeking	1	−0.075 **	-0.498 **	0.239 **	0.221	0.258	0.037	0.031	0.044	7.5
	2	−0.073 **	−0.493 **	0.233 **	0.215	0.252	0.036	0.030	0.043	7.4
Fantasy	1	0.052 **	0.047	0.239 **	0.221	0.258	0.002	−0.003	0.008	-
	2	0.053 **	0.073	0.233 **	0.215	0.252	0.004	−0.002	0.009	-
Problem avoidance	1	0.076 **	0.111 *	0.239 **	0.221	0.258	0.008	0.001	0.016	1.7
	2	0.075 **	0.104 *	0.233 **	0.215	0.252	0.008	0.001	0.015	1.6
Rationalization	1	0.051 **	0.163 *	0.239 **	0.221	0.258	0.008	0.003	0.014	1.7
	2	0.050 **	0.158 *	0.233 **	0.215	0.252	0.008	0.003	0.014	1.6
**Males**										
Problem solving	1	−0.076 **	−1.126 **	0.219 **	0.182	0.257	0.086	0.067	0.108	18.2
	3	−0.076 **	−1.127 **	0.213 **	0.175	0.251	0.086	0.067	0.109	18.5
Self-blame	1	0.092 **	1.500 **	0.219 **	0.182	0.257	0.138	0.111	0.170	29.2
	3	0.090 **	1.468 **	0.213 **	0.175	0.251	0.132	0.105	0.163	28.5
Help seeking	1	−0.065 **	−0.405 **	0.219 **	0.182	0.257	0.027	0.016	0.039	5.6
	3	−0.066 **	−0.419 **	0.213 **	0.175	0.251	0.028	0.017	0.040	6.0
Fantasy	1	0.065 **	−0.057	0.219 **	0.182	0.257	−0.004	−0.019	0.011	-
	3	0.065 **	−0.034	0.213 **	0.175	0.251	−0.002	−0.017	0.013	-
Problem avoidance	1	0.078 **	−0.002	0.219 **	0.182	0.257	−0.000	−0.017	0.017	-
	3	0.078 **	0.010	0.213 **	0.175	0.251	0.001	−0.016	0.018	-
Rationalization	1	0.048 **	0.143	0.219 **	0.182	0.257	0.007	−0.004	0.018	-
	3	0.048 **	0.140	0.213 **	0.175	0.251	0.007	−0.004	0.018	-
**Females**										
Problem solving	1	−0.083 **	−0.831 **	0.245 **	0.223	0.267	0.069	0.059	0.081	13.7
	3	−0.083 **	−0.836 **	0.242 **	0.220	0.263	0.069	0.059	0.080	13.9
Self-blame	1	0.100 **	1.273 **	0.245 **	0.223	0.267	0.127	0.113	0.142	25.2
	3	0.098 **	1.261 **	0.242 **	0.220	0.263	0.124	0.110	0.138	24.8
Help seeking	1	−0.077 **	−0.519 **	0.245 **	0.223	0.267	0.040	0.032	0.048	7.9
	3	−0.076 **	−0.517 **	0.242 **	0.220	0.263	0.039	0.032	0.047	7.9
Fantasy	1	0.047 **	0.095	0.245 **	0.223	0.267	0.005	−0.001	0.010	-
	3	0.048 **	0.111	0.242 **	0.220	0.263	0.005	−0.000	0.011	-
Problem avoidance	1	0.074 **	0.148 *	0.245 **	0.223	0.267	0.011	0.003	0.020	2.2
	3	0.074 **	0.138 *	0.242 **	0.220	0.263	0.010	0.002	0.019	2.0
Rationalization	1	0.051 **	0.170 *	0.245 **	0.223	0.267	0.009	0.002	0.015	1.7
	3	0.051 **	0.173 *	0.242 **	0.220	0.263	0.009	0.002	0.016	1.8

* *p* < 0.05; ** *p* < 0.001; a: Effect of CM (childhood maltreatment) on mediators, b: Effect of mediators on depressive symptoms. Model 1: single-factor analysis. Model 2: adjusted for age, gender, urban/rural, school, only child status, parents’ education level, economic status of family. Model 3: adjusted for age, urban/rural, school, only child status, parents’ education level, economic status of family. LLCI = lower limit confidence interval, ULCI = upper limit confidence interval.

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
