# Peer review of "Mediating Effects of Specific Types of Coping Styles on the Relationship between Childhood Maltreatment and Depressive Symptoms among Chinese Undergraduates: The Role of Sex"

_ijerph, 2020, doi:10.3390/ijerph17093120_

Round 1
Reviewer 1 Report
Dear Authors please read carefully my questions and proceed with the revision of the paper.
Lines 37-38: non clear. "Which" is probably incorrect.
Line 53: non space before reference. This error also in the rest of the document.
Line 68: please cite also the following reference: Tremolada, M., Bonichini, S., Basso, G., Pillon, M. (2015). Coping with pain in children with leukemia, International Journal of Cancer Research and Prevention, 8, 451-466.
Methodology: Please give more information about the measures adopted in the study. Which is the original version. The version used was standardized for the population of the study? Was it verifies that the respondents had knowledge about maltreatment, neglect, and emotional abuse?
No information are given about participants except for gender. Other socio-demographical information should be reported. Procedure: please explain procedure, how do you find your study population? How do you contact the respondents? How were administered the measures?
Table 4: p should be indicated in lowercase. Notes should be presented following the authors guidelines
Author Response
Lines 37-38: non clear. "Which" is probably incorrect.
Response: Thank you for this valuable suggestion. We have changed the statement of “Childhood maltreatment, including abuse or neglect experiences before 18 years old which may bring about actual or potential harm to the child’s health, survival, development or dignity.” to “Childhood maltreatment is behavior toward a person under 18 years of age,including abuse or neglect experiences, that results in actual or potential harm to the child’s health, survival, development or dignity and is perpetrated by a person of responsibility, trust, or power in that child’s life.” in the manuscript, lines 37-40.
Line 53: non space before reference. This error also in the rest of the document.
Response: Thank you very much for your reminder, we have inserted spaces before all reference in the manuscript.
Line 68: please cite also the following reference: Tremolada, M., Bonichini, S., Basso, G., Pillon, M. (2015). Coping with pain in children with leukemia, International Journal of Cancer Research and Prevention, 8, 451-466.
Response: We have read the article of Tremolada et al, and put it as reference 17.
Methodology: Please give more information about the measures adopted in the study. Which is the original version. The version used was standardized for the population of the study? Was it verifies that the respondents had knowledge about maltreatment, neglect, and emotional abuse?
Response: In the present study, depressive symptoms were evaluated using the Center for Epidemiologic Studies-Depression (CES-D) scale, a well-validated measure with reliability in ethnically diverse populations that is widely used in epidemiological studies [1]. In recent years, Li et al have used CES-D to evaluate the level of depressive symptoms in Chinese, the Cronbach's α coefficient were 0.895 [2], 0.941 [3] and 0.953 [4], respectively. We have added the statement of “the Chinese version of the questionnaire was widely used in Chinese populations and had good reliability and validity [22]” in the manuscript, lines 123-124.
Coping styles were assessed using the Coping style questionnaire (CSQ), which was created based on the Chinese culture and language [5]. And the CSQ has been widely used in studies of coping styles among Chinese undergraduates [5-6]. In Tang et al’ s study [6], Cronbac’s α values for the six subscales were as follows: problem-solving, 0.77; self-blaming, 0.72; help-seeking, 0.73; fantasizing, 0.72; avoidance, 0.70; and rationalizing, 0.72. We have added the statement of “Coping styles were assessed using the Coping style questionnaire (CSQ), which was created based on the Chinese culture and language.” in the manuscript, lines 131-132.
Childhood maltreatment was evaluated using the Child Trauma Questionnaire (CTQ) [7], a widely used 28-item measure that assesses five different forms of childhood trauma (physical abuse, sexual abuse, emotional abuse, physical neglect, and emotional neglect). The CTQ has been translated and validated into Chinese version and shown to have good reliability and validity(Cronbach’s α coefficient: 0.77) [8]. Wang et al. used CTQ to evaluate childhood maltreatment, and the Cronbach's α coefficient were 0.81 [9] and 0.78 [10].
No information are given about participants except for gender. Other socio-demographical information should be reported. Procedure: please explain procedure, how do you find your study population? How do you contact the respondents? How were administered the measures?
Response: More female students (65.5%) participated in the study, and the respondents were distributed in the Anhui Medical College (4206 students) and Anqing Medical College (3437). Among all participants, 63.2% came from rural areas and 36.8% from urban areas. Approximately three-quarters (78.5%) of participants reported that they were the only child and 21.5% having at least one sibling. Among the study population, 36.5% reported poor family financial status, 60.1% moderate, and 3.3% good. 30.3% of the participants reported that their father’ s education level was less than junior middle school, 44.9% junior middle school, 17.2% high school and 7.5% college or higher (the corresponded mother’ s education level were 50.9%, 32.8%, 11.5% and 4.8%). We have added the statement of “More female students (65.5%) participated in the study, and the respondents were distributed in the Anhui Medical College (4206 students) and Anqing Medical College (3437). Among all participants, 63.2% came from rural areas and36.8% from urban areas. Approximately three-quarters (78.5%) reported that they were the only child and 21.5% having at least one sibling.” in the manuscript, lines 114-117.
We have added the statement of “In the present study, investigators entered schools and classrooms to conduct surveys and did not occupy student class hours. The investigation was located in the classroom. Teachers were invited to take charged for maintaining discipline, but not participate in the survey. The teachers sent the link of the electronic questionnaire to each student’s mobile phone, and then the students began to complete the questionnaire. The investigators introduced the objective of the survey and the questions to be aware of when filling out questionnaires, and emphasized the principles of anonymity and confidentiality participation. Investigators answered questions from students on the spot.” in the manuscript, lines 102-110.
Table 4: p should be indicated in lowercase. Notes should be presented following the authors guidelines
Response: Thanks, we have replaced “P” with lowercase “p” in the manuscript, line 24 and table2-4.
[1] Roberts, R.E. Reliability of the CES-D Scale in different ethnic contexts. Psychiatry Res. 1980, 2, 125-134. https://doi.org/10.1016/0165-1781(80)90069-4
[2] Gu, Z.H.; Qiu, T.; Tian, F.Q.; Yang, S.H.; Wu, H. Perceived organizational support associated with depressive symptoms among petroleum workers in China: A cross-sectional study. Psychol. Res. Behav. Manag. 2020, 13, 97-104. https://doi.org/10.2147/PRBM.S232635
[3] Wei, Y.X.; Wang, X.T.; Zhang, J.; Yao, Z.Y.; Liu, B.P.; Jia, C.X. Psychometric properties of the psychological strain scales (PSS) in suicide attempters and community controls of rural China. J. Affect. Disord. 2020, 266, 753-759. https://doi.org/10.1016/j.jad.2020.01.105
[4] Zhang, H.; Shang, Z.; Wu, L.; Sun, Z.; Zhang, F.; Sun, L.; Zhou, Y.; Wang, Y.; Liu, W. Prolonged grief disorder in Chinese Shidu parents who have lost their only child. Eur. J. Psychotraumatol. 2020, 11, 1726071. https://doi.org/10.1080/20008198.2020.1726071
[5] Xiao, J.; Xu, X. The research of reliability and validity of the “Coping Style Questionnaire”. China Mental Health 1996, 10, 164-168.
[6] Tang, W.; Dai, Q. Depressive symptoms among first-year Chinese undergraduates: The roles of socio-demographics, coping style, and social support. Psychiatry Res. 2018, 270, 89-96. https://doi.org/10.1016/j.psychres.2018.09.027
[7] Bernstein, D.P.; Ahluvalia, T.; Pogge, D.; Handelsman, L. Validity of the Childhood Trauma Questionnaire in an adolescent psychiatric population. J. Am. Acad. Child Adolesc. Psychiatry 1997, 36, 340-348. https://doi.org/10.1097/00004583-199703000-00012
[8] Zhao, X.F.; Zhang, Y.L.; Li, L.F.; Zhou, Y.F.; Li, H.Z.; Yang, S.C. Reliability and validity of the Chinese version of Childhood Trauma Questionnaire. Chin. J. Clin. Rehabil. 2005, 9, 105-107.
[9] Wang, W.; Wu, R.; Tang, H.; Wang, Y.; Liu, K.; Liu, C.; Zhou, L.; Liu, W.; Deng, X.; Pu, W. Childhood trauma as a mediator between emotional intelligence and recidivism in male offenders. Child Abuse. Negl. 2019, 93, 162-169. https://doi.org/10.1016/j.chiabu.2019.04.015
[10] Gong, J.; Wang, Y.; Liu, J.; Fu, X.; Cheung, E.F.C.; Chan, R.C.K. The interaction between positive schizotypy and high sensitivity C-reactive protein on response inhibition in female individuals. Psychiatry Res. 2019, 274, 365-371. https://doi.org/10.1016/j.psychres.2019.02.064
Reviewer 2 Report
The manuscript addresses the relationship between coping styles, childhood maltreatment, and depression, taking into account the role of sex. This is a very interesting topic with important conclusions for interventions. The sample is one of the strengths of this manuscript and results in highlights that some coping strategies have a meditational effect on the relationship between childhood maltreatment and depression.
However, the manuscript has some weakness that should be improved:
In general terms, one of the questions that should be addressed is related to the introduction. There are different variables such as childhood maltreatment, exposure to childhood maltreatment, childhood adversity experiences… they are not defined properly and moreover, authors assess childhood maltreatment. Therefore, the authors should be focused on childhood maltreatment and develop more in-depth that variable.
On the other hand depression in childhood and coping strategies should be better explained and, most important, link all the variables. For instance, authors should incorporate recent studies about comorbidity between childhood maltreatment and depression. And previous studies about coping strategies.
It also should be better justified the role of sex on these relationships.
The objective should be clear and placed before the hypotheses. Moreover, the hypotheses are too ambiguous. Should be better explained.
As for the Method, the procedure should be better explained as well as the sampling structure. We do not know how the information about participants’ age, urban/rural status, school, only child status, parental educational level (less than the junior middle, school, junior middle school, senior middle school, college, or higher), and the perceived economic status of the family (poor, moderate, or good).
One important question is that the authors assess childhood maltreatment from a retrospective perspective (participants were university students) and current depressive symptoms. This strategy has important limitations that should be addressed.
The discussion has to be more in-depth because it has the same questions as the introduction. It is too descriptive and there’s a lack of theory that helps to interpret the results.
Author Response
Reviewer 2
In general terms, one of the questions that should be addressed is related to the introduction. There are different variables such as childhood maltreatment, exposure to childhood maltreatment, childhood adversity experiences… they are not defined properly and moreover, authors assess childhood maltreatment. Therefore, the authors should be focused on childhood maltreatment and develop more in-depth that variable.
Response: Thank you for your suggestion, we have added the statement of “Childhood maltreatment is behavior toward a person under 18 years of age,including abuse or neglect experiences,that results in actual or potential harm to the child’s health, survival, development or dignity and is perpetrated by a person of responsibility, trust, or power in that child’s life” in the manuscript, lines 37-40.
We have added the statement of “childhood adversity (child abuse and neglect)” in the manuscript, page 2, line 3 and “childhood abuse experiences (sexual abuse, physical abuse, and negative home environment)” in the manuscript, line 62.
On the other hand depression in childhood and coping strategies should be better explained and, most important, link all the variables. For instance, authors should incorporate recent studies about comorbidity between childhood maltreatment and depression. And previous studies about coping strategies.
Response: Thanks, we have read some recent literature about the correlations between childhood maltreatment, coping styles, and depressive symptoms, and put the articles of Vallati et al, Petersen et al and Teseia et al as references 3, 12 and 13 respectively in the revised manuscript. We have added the statement of “Research by Vallati et al. also showed that high levels of emotional maltreatment and/or sexual maltreatment were significantly associated with severe depressive symptoms [3].” in the manuscript, lines 46-48.
It also should be better justified the role of sex on these relationships.
Response: We have added the statement of “Furthermore, Matud et al. have speculated that differences in the way women cope with stress could be related to their higher levels of psychological distress, symptoms of depression and anxiety compared with men [18].” in the manuscript, lines 84-86.
The objective should be clear and placed before the hypotheses. Moreover, the hypotheses are too ambiguous. Should be better explained.
Response: We have added the statement of “Hence, the current study aims to investigate the possible mediating roles of different types of coping styles in the relationship between childhood maltreatment and depressive symptoms among Chinese undergraduates and the sex difference in the mediated pathways.” in the manuscript, lines 89-91.
We have changed the statement of “sex differences affect the mediating role of coping styles in the association between childhood maltreatment and depressive symptoms” to “The coping styles as an mediating role are varied by sex in the association between childhood maltreatment and depressive symptoms.” in the manuscript, lines 94-96.
As for the Method, the procedure should be better explained as well as the sampling structure. We do not know how the information about participants’ age, urban/rural status, school, only child status, parental educational level (less than the junior middle, school, junior middle school, senior middle school, college, or higher), and the perceived economic status of the family (poor, moderate, or good).
Response: More female students than male students participated in the study (65.5% vs. 34.5%), and the respondents were distributed in the Anhui Medical College (4206 students) and Anqing Medical College (3437). Among all participants, 63.2% came from rural areas, while 36.8% came from urban areas. Just over half of participants (78.5%) reported that they were the only child, while 21.5% reported having at least one sibling.36.5% of the participants reported poor family financial status, 60.1% were moderate, and 3.3% were good. 30.3% of the participants reported that their father’ s education level was less than junior middle school, 44.9% junior middle school, 17.2% high school and 7.5% college or higher (mother’ s education level: 50.9% less than junior middle school, 32.8% junior middle school, 11.5% high school and 4.8% college or higher). We have added the statement of “More female students than male students participated in the study (65.5% vs. 34.5%), and the respondents were distributed in the Anhui Medical College (4206 students) and Anqing Medical College (3437 students). Among all participants, 63.2% came from rural areas, while 36.8% came from urban areas. Just over half of participants (78.5%) reported that they were the only child, while 21.5% reported having at least one sibling.” in the manuscript, lines 114-118.
One important question is that the authors assess childhood maltreatment from a retrospective perspective (participants were university students) and current depressive symptoms. This strategy has important limitations that should be addressed.
Response: Thanks for a good suggestion. We have added the statement of “childhood maltreatment was assessed by a retrospective questionnaire, and thus recall bias cannot be avoided in this study” in the manuscript, lines 324-326.
The discussion has to be more in-depth because it has the same questions as the introduction. It is too descriptive and there’s a lack of theory that helps to interpret the results.
Response: We have added the statement of “The theoretical models proposed by Nusslock et al. [38-39] showed that the biological and psychosocial changes caused by childhood maltreatment are predictors of many negative outcomes in adulthood (eg., health status, drug abuse, psychological status). These biopsychosocial models show that, in particular, avoidant emotion-focused coping strategies parallel and interact with other biological (eg., impaired immune function) and psychosocial (eg., problematic health behaviors) pathways that affect adult diseases. One explanation for this is that childhood maltreatment may increase the sensitivity of brain regions involved in stress responses, inhibitory control, and reward responses [38]. These neurobiological and psychological changes in turn affect the cognitive assessment of threats and the response to perceived threats. Therefore, people who are exposed to ACEs are more likely to experience stressful situations, but less likely to develop effective coping styles, which might result in more psychological problems.” in the manuscript, lines 264-275.
Reviewer 3 Report
Authors present an interesting article about mediatory role of coping styles in the relationship between childhood maltreatment and depressive symptoms among Chinese undergraduates, including the role of sex. The manuscript is well written and clear, the statistical analyses are appropriate, the rationale behind the mediatory process is convincing, and the results are correctly supported. A total of participants is impressive
I would like to ask the Authors to address some points in order to improve the paper.
Line 127: I think it is enough to write „ranged from 1 = „never true” to 5 = „very often true”
Line 140: Did Authors check the the assumption for Pearson’s correlation? – I especially refer myself to the outliers and to approximate normal distribution. It would be good to report the statistics relating to both assumptions
Line 148: Which mediatory model did Authors use? It would be helpful to mention it
Author Response
Line 127: I think it is enough to write „ranged from 1 = „never true” to 5 = „very often true”
Response: We have changed the statement of “Response scores ranged from 1 = ‘never true,’ 2 = ‘rarely true,’ 3 = ‘sometimes true,’ 4 = ‘often true,’ and 5 = ‘very often true.’” to “Response scores ranged from 1 = ‘never true,’ to 5 = ‘very often true.’” in the manuscript, page 3, lines 148-149.
Line 140: Did Authors check the the assumption for Pearson’s correlation? – I especially refer myself to the outliers and to approximate normal distribution. It would be good to report the statistics relating to both assumptions
Response: The variables involved in this article are all multiple-choice items, and thus no outliers in this electronic questionnaire survey. The present article referred to the research of Titelius et al. [1-3] and conducted Pearson correlations to explore the association between childhood maltreatment, coping styles and depressive symptoms. However, with your valuable suggestions, a scatter plot was drawn and shown a slight skewed distribution. Therefore, we chose Spearman correlation to re-analyze the association between childhood maltreatment, coping style and depressive symptoms (see table 2).
Line 148: Which mediatory model did Authors use? It would be helpful to mention it
Response: PROCESS program of mediation (model 4) was used to perform a multiple mediation analysis. And we have changed the statement of “Then, PROCESS program of mediation was used to perform a multiple mediation analysis” to “Then, PROCESS program of mediation (model 4) was used to perform a multiple mediation analysis” in the manuscript, line 167.
[1] Titelius, E.N.; Cook, E.; Spas, J.; Orchowski, L.; Kivisto, K.; O'Brien, K.H.M.; Frazier, E.; Wolff, J.C.; Dickstein, D.P.; Kim, K.L.; Seymour, K. Emotion dysregulation mediates the relationship between child maltreatment and non-suicidal self-snjury. J. Aggress. Maltreat. Trauma. 2018, 27, 323-331. https://doi.org/10.1080/10926771.2017.1338814
[2] Kang, N.; Jiang, Y.; Ren, Y.; Gong, T.; Liu, X.; Leung, F.; You, J. Distress intolerance mediates the relationship between child maltreatment and nonsuicidal self-Injury among Chinese adolescents: A three-wave longitudinal study. J. Youth. Adolesc. 2018, 47, 2220-2230. https://doi.org/10.1007/s10964-018-0877-7
[3] Hong, F.; Tarullo, A.R.; Mercurio, A.E.; Liu, S.; Cai, Q.; Malley-Morrison, K. Childhood maltreatment and perceived stress in young adults: The role of emotion regulation strategies, self-efficacy, and resilience. Child Abuse Negl. 2018, 86, 136-146. https://doi.org/10.1016/j.chiabu.2018.09.014
Round 2
Reviewer 2 Report
The manuscript has deeply improved.
My last suggestion is to organize the discussion based on the objectives and the hypothesis instead of the kind of analysis. This is more accurate to the scientific method. It is supposed that the analytical plan has the purpose of achieving the objectives.
Author Response
My last suggestion is to organize the discussion based on the objectives and the hypothesis instead of the kind of analysis. This is more accurate to the scientific method. It is supposed that the analytical plan has the purpose of achieving the objectives.
Response: Thank you for this valuable suggestion. We have added the statement of “the current study aims to identify the relationships between childhood maltreatment, coping styles, and depressive symptoms and investigate the possible mediating roles of different types of coping styles in the relationship between childhood maltreatment” in the manuscript, lines 89-81, “The first objective of this study was to identify the relationships between childhood maltreatment, coping styles, and depressive symptoms.” in the manuscript, lines 222-223 and “In fact, another purpose of this research was to estimate the possible mediating role of various types of coping style in the link between childhood maltreatment and depressive symptoms among Chinese undergraduates.” in the manuscript, lines 253-255.